# Comparison of Interleaved Boost Converter and Two-Phase Boost Converter Characteristics for Three-Level Inverters

**Eiichi Sakasegawa** [1,*] **, Rin Chishiki** [2] **, Rintarou Sedutsu** [1] **, Takumi Soeda** [2] **, Hitoshi Haga** [2] **and Ralph Mario Kennel** [3]

1   National Institute of Technology, Kagoshima College, 1460-1, Kirishima, Kagoshima 899-5193, Japan
2   Department of Electrical, Electronics and Information Engineering, Nagaoka University of Technology, 1603-1, Kamitomiokamachi, Nagaoka, Niigata 940-2188, Japan
3   TUM School of Engineering and Design Department of Energy and Process Engineering, Technical University of Munich, Arcisstrasse 21, 80333 Munich, Germany
*   Correspondence: sakasegw@kagoshima-ct.ac.jp

**Abstract:** A boost converter is used in various applications to obtain a higher voltage than the input voltage. One of the current main circuit systems for hybrid electric vehicles (HEVs) is a combination of a two-phase boost converter (parallel circuit) and a three-phase two-level inverter. In this study, we focus on the boost converter to achieve even higher efficiency and propose an interleaving scheme for a boost converter suitable for a three-level inverter (series circuit). The series circuit has two capacitors connected in series and makes it suitable as a power supply for a three-level inverter. We analyze the input current ripple of the series and parallel circuit in order to show the superiority of the series circuit. Furthermore, we propose a novel output voltage control strategy using an optimal regulator, namely a Linear Quadratic Regulator (LQR), for the series circuit. As a result, we found the input current ripple of the series circuit is smaller than the parallel circuit and demonstrated the superiority of the series circuit. The simulation and experimental results show the effectiveness of the proposed interleaving scheme and optimal regulator.

**Keywords:** EV; HEV; three-level inverter; two-phase boost converter; interleaved boost converter; input current ripple; optimal regulator

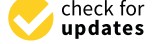



## 1. Introduction

As a countermeasure to recent environmental problems, electrification has been widely promoted in the transportation sector, including automobiles [1–3]. Voltage-source three-phase inverters are widely used for the variable speed drives of the main motors for hybrid electric vehicles (HEVs), and the DC power is supplied by a battery through a boost converter [4]. In the boost converter system, there is a strong need to downsize the passive elements that make up the bulk of the system. Many studies have been conducted to miniaturize the passive components. One of the main methods with respect to circuit topology for HEVs is to apply a two-phase boost converter [5,6] As for the control strategy, there are methods to reduce the ripple of the battery input current by employing an interleaving method using a two boost converter [7], to reduce the DC link capacitor current by cooperative control of the boost converter and inverter to downsize the smoothing capacitor [8], and to suppress voltage fluctuations by load current feed-forward control [9].

Multilevel inverters have lower output voltage harmonics due to the increase in the voltage levels with lower $dv/dt$ compared with conventional two-level inverters, which results in a decrease in the switching losses. Due to the advantages, the motor drive system is able to obtain a higher efficiency. Therefore, multilevel inverters are promising power converters for the main motors of future automobiles including HEVs [10,11]. However, balancing the control of capacitor voltages is required. For the capacitor voltage balancing

control method, a method to devise the switching pattern of the inverter is used [12–14]. Reference [15] reports a method in which a boost converter is connected to a three-level inverter and the capacitor voltage is balanced and controlled by the boost converter. The series connected boost converter of this method not only performs the boost operation but also balances the capacitors. This method achieves coordinated control of the boost converter and three-level inverter. However, in [15], a PWM scheme to reduce the input current ripple was not discussed.

Therefore, this paper proposes an interleaving scheme for series connected boost converters that can be applied to three-level inverters [16]. The effectiveness of the proposed interleaving scheme is clarified by comparing the input current ripple characteristics with those of the parallel-connection-type interleaving scheme. Furthermore, for the output voltage control method, we propose a control method using an optimal regulator as a method that is highly effective in suppressing output voltage fluctuations and has excellent extendibility when connected to a three-level inverter. The effectiveness of the proposed optimal regulator is clarified by comparing its response with that of PI control.

The novelty of the paper is the interleaved boost converter topology for a main motor drive system in HEVs in which an NPC inverter is applied, and its control strategy. The balancing control of the input voltages is required when the NPC inverter is applied for the HEVs because the NPC inverter has three input terminals and the neutral point potential varies due to the imbalance of the terminal voltages of the load. The balancing control causes a decrease in the inverter voltage utilization factor, which results in a decrease in the system efficiency. The proposed interleaved boost converter (series circuit) is capable of input voltage balancing control for the NPC inverter and improves the conversion efficiency compared with that of the conventional boost converter (parallel circuit). Therefore, this paper contributes an improvement in the efficiency of the motor drive system. This paper is organized as follows. Chapter 2 describes the two circuit configurations, and Chapter 3 derives a theoretical expression for the input current ripple. Chapter 4 describes the control method of the output voltage. Chapter 5 presents the results of simulations and experiments based on the derived theoretical value of the input current ripple. The response of the output voltage with the proposed controller is also presented. In addition, a comparison of the efficiency of the two circuits is presented. The validity of the theoretical equation for the input current ripple presented in this paper and the effectiveness of the interleaved scheme are demonstrated by simulation and experimental results.

## 2. Circuit Configuration

In this chapter, the series and parallel circuits are presented and the switching mode analysis in the interleaved scheme is presented. It is assumed that a three-level inverter is connected as the load of the boost converter. However, a resistor is connected as the load because this paper focuses on the input current ripple and output voltage characteristics due to boost operation.

### 2.1. Interleaving Scheme

Figures 1 and 2 show the parallel and series circuits, respectively. Figure 1 shows a configuration in which the boost converters are connected in parallel. Figure 2 shows a configuration in which the boost converters are connected in series. This paper proposes an interleaving method to generate the gate signals for series circuits. Figure 3 shows the principle of the interleaving method, where $D_p$ and $D_n$ are the duty ratios given to the switches ($S_1$ and $S_2$) in Figures 1 and 2, respectively. Figure 3a shows the case where $D \leq 0.5$ and Figure 3b shows the case where $D > 0.5$. As shown in Figure 3a, the controller modulates the command value using two carriers $f_{c1}$ and $f_{c2}$ with a phase difference of 180 degrees to generate the gate signals for the switches. Here, each switch, carrier, and duty ratio correspond to $f_{c1}$ and $D_p$ for $S_1$, and $f_{c2}$ and $D_n$ for $S_2$, respectively. Due to the interleaving method, the frequency of the input current ripple is twice the carrier frequency.

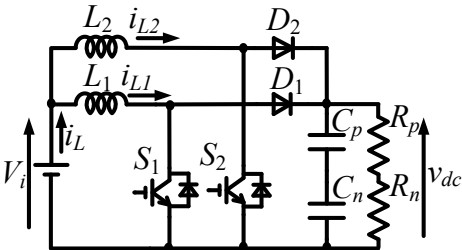

**Figure 1.** Parallel circuit.

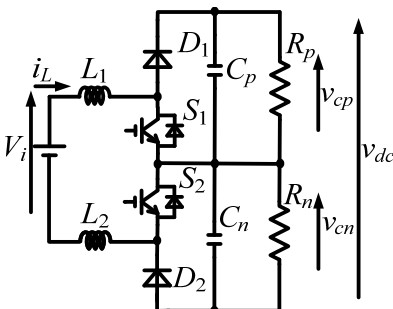

**Figure 2.** Series circuit.

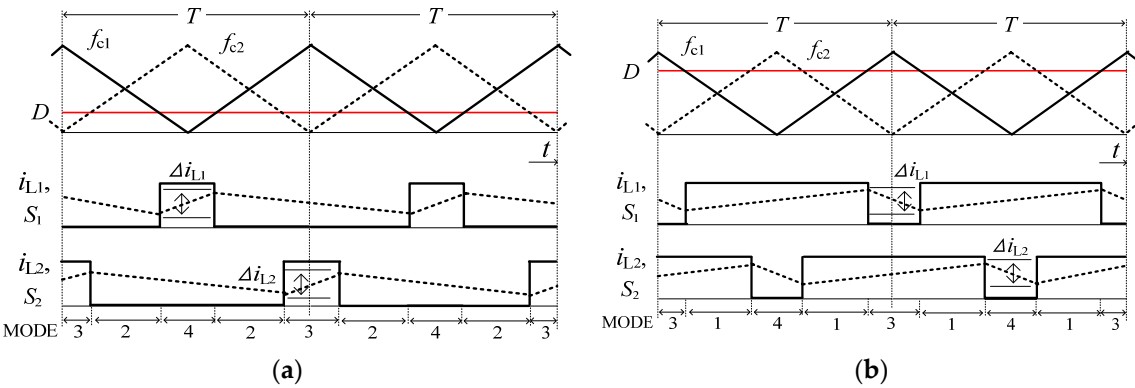

**Figure 3.** Carriers and switching modes of interleaving method. (**a**) $D \leq 0.5$; (**b**) $D > 0.5$.

### 2.2. Switching Modes

In the two circuits, four modes appear depending on the magnitude of $D$. The switching modes of the parallel and series circuits are shown in Figures 4 and 5, respectively. In both circuits, MODE1 in Figures 4a and 5a is a mode in which both $S_1$ and $S_2$ are on, MODE2 in Figures 4b and 5b is a mode in which both $S_1$ and $S_2$ are off, MODE3 in Figures 4c and 5c is a mode in which $S_1$ is off and $S_2$ is on, and MODE4 in Figures 4d and 5d is a mode in which $S_1$ is on and $S_2$ is off. For $D > 0.5$ in Figure 3b, magnetic energy is stored in both reactors with MODE 1 in Figures 4a and 5a and boosted with MODE 3 in Figures 4c and 5c and MODE 4 in Figures 4d and 5d alternately. The upper and lower capacitor voltages ($v_{cp}$, $v_{cn}$) of the series circuit are controlled separately by charging the upper capacitor with MODE 3 in Figure 5c and the lower capacitor with MODE 4 in Figure 5d.



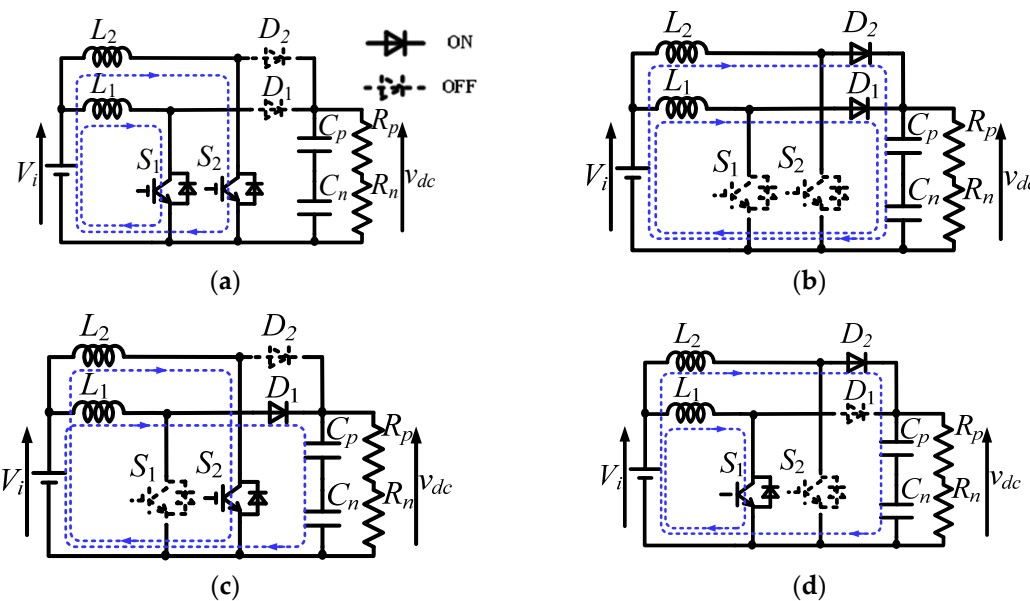

**Figure 4.** Switching modes of parallel circuit. (**a**) MODE1; (**b**) MODE2; (**c**) MODE3; (**d**) MODE4.

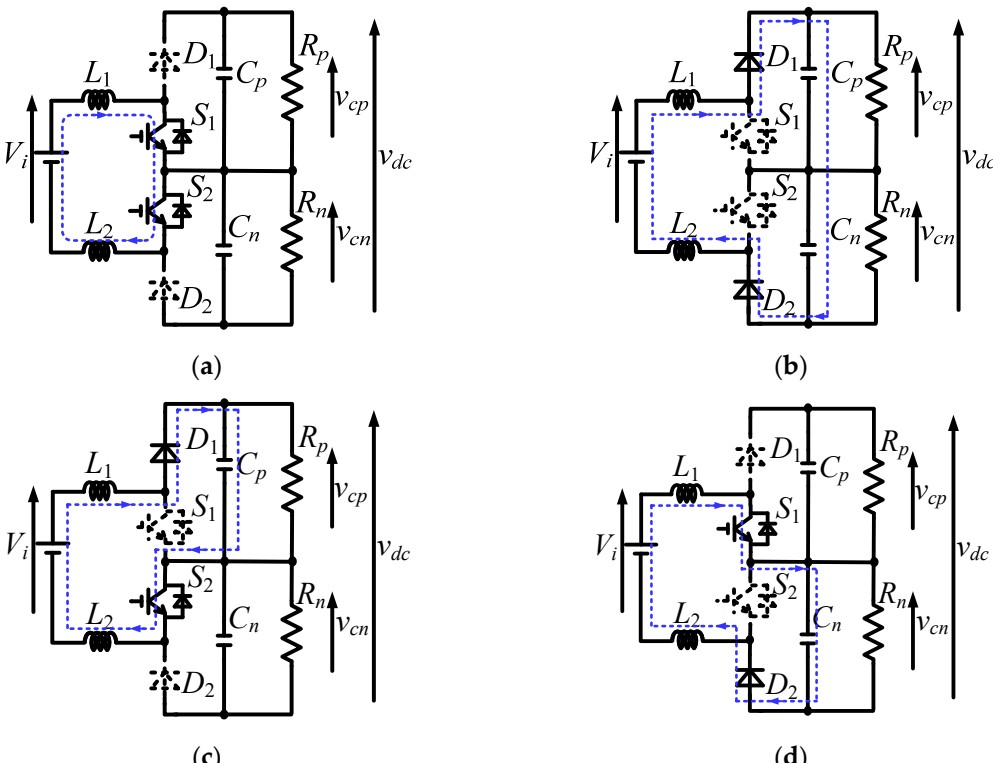

**Figure 5.** Switching modes of series circuit. (**a**) MODE1; (**b**) MODE2; (**c**) MODE3; (**d**) MODE4.

## 3. Input Current Ripple Analysis

This chapter derives a theory of the switching ripple $\Delta i_L$ of the input current $i_L$ for both circuits. As the operating mode of the input current, this chapter assumes a current continuous mode. For simplicity of analysis, each phase of the parallel circuit and the upper and lower of the series circuit are symmetrical. Each inductance value and duty ratio of the two switches are also the same. In other words, this chapter is analyzed with $L_1 = L_2 = L$ and $D_p = D_n = D$. This chapter also discusses the input current ripple considering the rated current of the reactor.

*3.1. Input Current Ripple in Parallel Circuits*

In a parallel circuit, each reactor is connected in parallel to the power supply, so the input current ripple $\Delta i_L$ is the sum of each reactor current ripple $\Delta i_{L1}$ and $\Delta i_{L2}$. Figure 3a shows the gate signal and each reactor current $i_{L1}$ and $i_{L2}$ when $D \leq 0.5$. When both $S_1$ and $S_2$ are off, each reactor current ripple $\Delta i_{L1}$ and $\Delta i_{L2}$ are as follows [7]:

$$\Delta i_{L1} = \frac{v_{dc} - V_i}{L}(0.5 - D)T \tag{1}$$

$$\Delta i_{L2} = \frac{v_{dc} - V_i}{L}(0.5 - D)T \tag{2}$$

where $\Delta i_{L1}$, $\Delta i_{L2} > 0$ as defined in Figure 3 due to $V_i < v_{dc}$.

Each reactor current has the same peak-to-peak value and a phase difference 180 degrees as shown in Figure 3a. The input current ripple $\Delta i_L$ becomes their sum, and considering $v_{dc} = V_i/(1 - D)$, the current ripple becomes Equation (3).

$$\Delta i_L = \Delta i_{L1} + \Delta i_{L2} = \frac{2V_i}{L}\frac{(0.5 - D)}{(1 - D)}DT \tag{3}$$

Figure 3b shows the gate signal and each reactor current when $D > 0.5$. In this case, when both $S_1$ and $S_2$ are on, each reactor current ripple $\Delta i_{L1}$ and $\Delta i_{L2}$ are as in Equation (4).

$$\Delta i_{L1} = \Delta i_{L2} = \frac{V_i}{L}(D - 0.5)T \tag{4}$$

The input current ripple $\Delta i_L$ is expressed as Equation (5).

$$\Delta i_L = \Delta i_{L1} + \Delta i_{L2} = \frac{2V_i}{L}(D - 0.5)T \tag{5}$$

*3.2. Input Current Ripple in Series Circuits*

In a series circuit, each reactor is connected in series with the input power supply, so the reactor value is doubled and the input current ripple $\Delta i_L$ is half that of one phase in a parallel circuit. In other words, the current ripple is half of Equation (1) when $D \leq 0.5$. Moreover, when $v_{dc} = V_i/(1 - D)$ is taken into account, Equation (6) is obtained.

$$\Delta i_L = \frac{V_i}{2L}\frac{(0.5 - D)}{(1 - D)}DT \tag{6}$$

On the other hand, the current ripple for $D > 0.5$ is half of Equation (4).

$$\Delta i_L = \frac{V_i}{2L}(D - 0.5)T \tag{7}$$

Based on the above equations, if the inductance values of both the parallel circuit and the series circuit are equal, the input current ripple of the series circuit is one-quarter that of the parallel circuit. Next, the input current ripple is discussed considering the rated current of the reactor. Since the parallel circuit is connected in parallel, the rated current of the reactor is half of the input current. On the other hand, the series circuit is series connected, so the rated current of the reactor is the same as the input current. Therefore, the rated current of the reactor in a series circuit is twice that of a parallel circuit. The volume and weight of the reactor are approximately determined by the product of the inductance value and the square of the reactor current ($LI^2$). If the inductance value of the series circuit is set to one-quarter of the parallel circuit, the input current ripple of the two circuits will be equal.

**4. Output Voltage Control**

This chapter describes the output voltage control method using PI control. After that, the output voltage control method using the optimal regulator (LQR) is described. There

are two advantages of applying LQR to multilevel inverters. First, LQR allows a fast load response by using optimal control input. In Reference [8], it is demonstrated that the fast load response of the controller is capable of downsizing the smoothing capacitor. The series circuit is designed to be applied to a three-level inverter, which requires a larger smoothing capacitor than a two-level inverter in order to balance the neutral point potential in the DC link voltage. Therefore, LQR can contribute to the reduction in the smoothing capacitor capacity of the three-level inverter. Next, when the number of state variables increases by connecting a boost converter and a three-level inverter, LQR can collectively control the state variables of the extended system due to its excellent extendibility.

*4.1. PI Control*

Figures 6 and 7 show the control block diagram of each boost converter circuit. In Figure 6, the reactor current ($i_{L1}$, $i_{L2}$) and output voltage ($v_{dc}$) are obtained by sensors, and the deviation between the voltage command value and the actual voltage is input to the PI controller to obtain the input current command. The deviation between the current command value and the actual current is then input to the PI controller to calculate the duty ratio of the boost converter, which is modulated by a carrier wave to generate a gate signal. In Figure 7, the capacitor voltages ($v_{cp}$, $v_{cn}$) and reactor current ($i_L$) are obtained by sensors, and the duty ratio is calculated as in Figure 6. The difference between Figures 6 and 7 is that a control system for the neutral point potential (NPP) is added to control the balance between the upper and lower capacitor voltages. Neutral point potential $v_n$ is defined as the following equation:

$$v_n = \frac{v_{cn} - v_{cp}}{2} \tag{8}$$

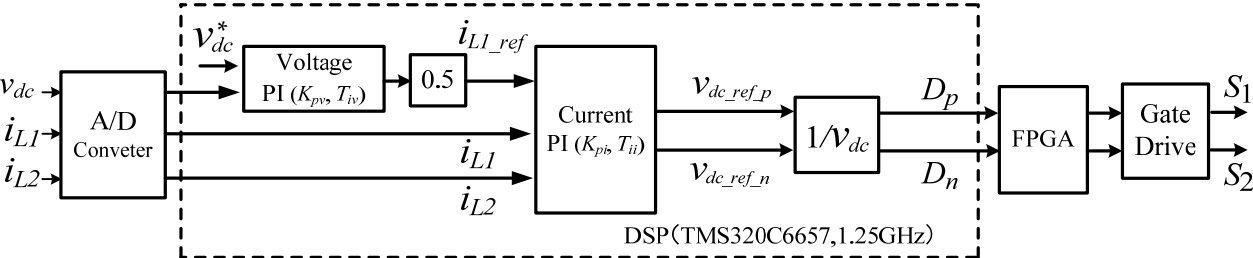

**Figure 6.** Configuration of parallel circuit with PI controller.

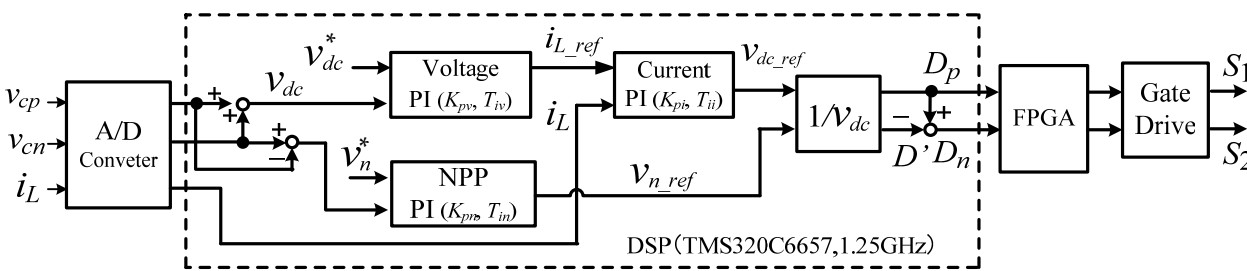

**Figure 7.** Configuration of series circuit with PI controller.

The command value of the neutral point potential is set to $v_n^* = 0$. The current PI controller outputs $v_{dc\_ref}$ so that $D_p$ equals $D_n$ when the neutral point potential is 0 V. The NPP PI controller outputs $v_{n\_ref}$ according to the error of the neutral point potential from the command value to realize the balance control. Defining the difference between $D_p$ and $D_n$ as $D' = D_p - D_n$, the NPP PI controller corrects $D_n$ with $D'$ when the neutral point potential varies.

*4.2. Output Voltage Control Using Optimal Regulator*

　　The state equation in the current continuous mode of the series circuit shown in Figure 2 can be expressed by the following equation when the state averaging method is used, where the output time of each MODE is $T_1 = (D_p + D_n - 1)\,T$ for MODE1, $T_3 = (1 - D_p)\,T$ for MODE3, and $T_4 = (1 - D_n)\,T$ for MODE4, for $D > 0.5$ shown in Figure 3b.

$$\dot{x}(t) = Ax(t) + Bu(t) = f(x(t), u(t)) \tag{9}$$

$$y(t) = Cx(t) \tag{10}$$

$$x(t) := [i_L\ v_{cp}\ v_{cn}]^T, y(t) = \begin{bmatrix} 0 & 1 & 0 \\ 0 & 0 & 1 \end{bmatrix} x(t) \tag{11}$$

$$A = \begin{bmatrix} 0 & -\frac{\overline{D_p}}{L} & -\frac{\overline{D_n}}{L} \\ \frac{\overline{D_p}}{C} & -\frac{1}{CR} & 0 \\ \frac{\overline{D_n}}{C} & 0 & -\frac{1}{CR} \end{bmatrix}, B = \begin{bmatrix} 1/L \\ 0 \\ 0 \end{bmatrix} \tag{12}$$

where $u(t) := V_i$, $\overline{D_p} = 1 - D_p$, $\overline{D_n} = 1 - D_n$, $C = C_p = C_n$, $R = R_p = R_n$. From Equations (9)–(12), the state Equation (9) is a nonlinear system. In this paper, the state equation is designed to be linearized around the equilibrium point. The equilibrium point is set as $x(t) = X$, $u(t) = U$. Assuming that the variation in the equilibrium point is sufficiently slower than the carrier period, a small signal model is obtained as follows:

$$\Delta\dot{x}(t) = \Delta A \Delta x(t) + \Delta B \Delta u(t) \tag{13}$$

$$\Delta y(t) = \Delta C \Delta x(t) \tag{14}$$

$$\Delta A = \left[\frac{\partial f(X, U)}{\partial x(t)}\right]^T = \begin{bmatrix} 0 & -\frac{\overline{D_p}}{L} & -\frac{\overline{D_n}}{L} \\ \frac{\overline{D_p}}{C} & -\frac{1}{CR} & 0 \\ \frac{\overline{D_n}}{C} & 0 & -\frac{1}{CR} \end{bmatrix} \tag{15}$$

$$\Delta B = \left[\frac{\partial f(X, U)}{\partial u(t)}\right]^T = \begin{bmatrix} -\frac{V_{cp}}{L} & -\frac{V_{cn}}{L} \\ \frac{I_L}{C} & 0 \\ 0 & \frac{I_L}{C} \end{bmatrix} \tag{16}$$

$$\Delta C = \begin{bmatrix} 0 & 1 & 0 \\ 0 & 0 & 1 \end{bmatrix} \tag{17}$$

$$x(t) := X + \Delta x(t), \Delta x(t) := [\Delta i_L\ \Delta v_{cp}\ \Delta v_{cn}]^T$$
$$u(t) := U + \Delta u(t), \Delta u(t) := [\Delta \overline{D_p}\ \Delta \overline{D_n}]^T$$
$$X := [I_L\ V_{cp}\ V_{cn}]^T, U := [\overline{D_p}\ \overline{D_n}]^T$$

where $\overline{D_p} = \overline{D_n} = \overline{D}$, and the value of the equilibrium point is Equation (18).

$$\overline{D} = \frac{V_i}{V_{dc}}, \ I_L = \frac{V_i}{2R\overline{D}^2}, \ V_{cp} = V_{cn} = \frac{V_i}{2\overline{D}} \tag{18}$$

　　Equation (13) is expanded as the servo system so that the controlled variables $v_{cp}$ and $v_{cn}$ are able to follow the step response. The state equation of the expanded system with additional state variable $\omega(t)$, which is the integral of the deviation $e(t)$ between the controlled variables $v_{cp}$, $v_{cn}$ and their reference values, is expressed as the following equations:

$$\dot{\tilde{x}}_e(t) = \Delta A_e \tilde{x}_e(t) + \Delta B_e \tilde{u}_e(t) \tag{19}$$

$$\widetilde{u}_e(t) = F\widetilde{x}_e(t) \tag{20}$$

$$\Delta A_e = \begin{bmatrix} \Delta A & O \\ -\Delta C & O \end{bmatrix}, \quad \Delta B_e = \begin{bmatrix} \Delta B \\ O \end{bmatrix}, \quad F = [K\ G] \tag{21}$$

$$\widetilde{x}_e(t) := \begin{bmatrix} x(t) \\ \omega(t) \end{bmatrix} \tag{22}$$

$$u(t) = Kx(t) + G\omega(t), \quad \omega(t) := \int_0^t e(t)dt \tag{23}$$

Figure 8 shows the configuration of the derived controller. The optimal regulator is used to determine the gains $K$ and $G$ in Equation (23). The evaluation function is expressed as the following:

$$J = \int_0^\infty \left( x(t)^T Q x(t) + u(t)^T R u(t) \right) dt \tag{24}$$

where $Q$ is the weight coefficient matrix for each state variable and $R$ is the weight coefficient matrix for the magnitude of the control input.

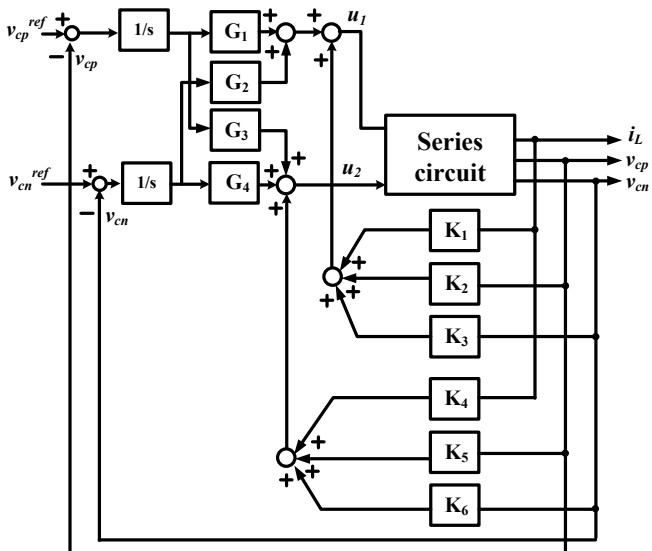

**Figure 8.** Configuration of series circuit with LQR.

## 5. Experimental Verification

### 5.1. PI Control Design

Figure 9 shows the experimental setup. Table 1 shows the main circuit parameters. The parameters of the PI controller were designed using Bode diagrams and simulations. The response speed of the output voltage was set to be about 0.1 s for a 50 V step change in the command value. Simulation of the circuit shown in Figure 7 with the parameters shown in Table 1 resulted in a control bandwidth of 500 rad/s for the current control system, which allows the system to boost voltage stably. The control bandwidth of the voltage control system is 1/10 of that of the current control system, and the control bandwidth of the neutral point potential PI control is less than 1/2 of that of the voltage control system. Table 2 shows the control parameters.

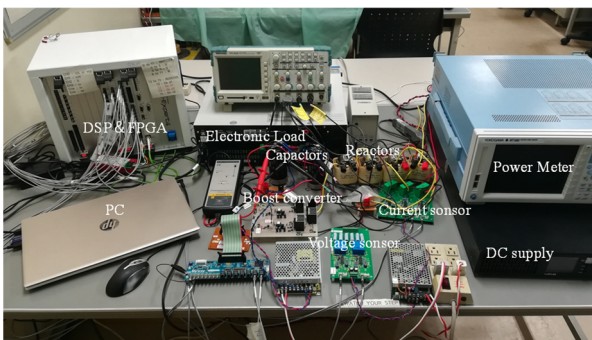

**Figure 9.** Experimental setup.

**Table 1.** Circuit conditions.

| | |
|---|---|
| Input voltage $V_i$ | 100 V |
| Carrier frequency $f_{c1}, f_{c2}$ | 10 kHz |
| IGBT<br>rated voltage<br>rated current | FGH40T120SMD<br>1200 V<br>40 A |
| Diode<br>rated voltage<br>rated current | FEP16DT<br>600 V<br>16 A |
| DC reactor<br>inductance $L_1, L_2$<br>rated current<br>core<br>resistance | -<br>1.8 mH<br>9 A<br>silicon steel plate<br>68.6 m$\Omega$ |
| Capacitanc $C_p, C_n$ | 1500 uF |

**Table 2.** Parameters of PI controllers.

| Parameter | Value |
|:---:|:---:|
| $K_{pi}$ | 4 |
| $T_{ii}$ | 4 ms |
| $K_{pv}$ | 0.075 |
| $T_{iv}$ | 40 ms |
| $K_{pn}$ | 2 |
| $T_{in}$ | 0.1s |

*5.2. Optimal Regulator Design*

In this paper, the weight coefficients in Equation (24), which are necessary to calculate the gain of LQR, are designed through simulations. The output voltage response is set to be about 0.1 s as well as that of PI control, and the weight coefficient matrices $Q$ and $R$ of the LQR are set as $Q$ = diag [$q_1$, $q_2$, $q_3$, $q_4$, $q_5$] and $R$ = diag [$r_1$, $r_2$], in which diag denotes the diagonal matrix. Each coefficient corresponds to $q_1$:$i_L$, $q_2$:$v_{cp}$, $q_3$:$v_{cn}$, $q_4$:$\Delta v_{cp}$, $q_5$:$\Delta v_{cn}$, $r_1$:$\overline{D_p}$, $r_2$:$\overline{D_n}$.

Figure 10 shows the simulated results when the weights are changed. Here, the weight coefficients for the inputs were set to $R$ = diag [1, 1]. Comparing Figure 10a,b, it can be seen that the larger the weight on the deviation, the faster the response. Comparing Figure 10c,d, it can be seen that the response can be made faster by increasing the weight of $\Delta v_{cn}$. Simulated results show that the desired response can be obtained when Figure 10e is used. For gain calculation, the "Arimoto-Potter method" is used. Table 3 shows the gains designed.

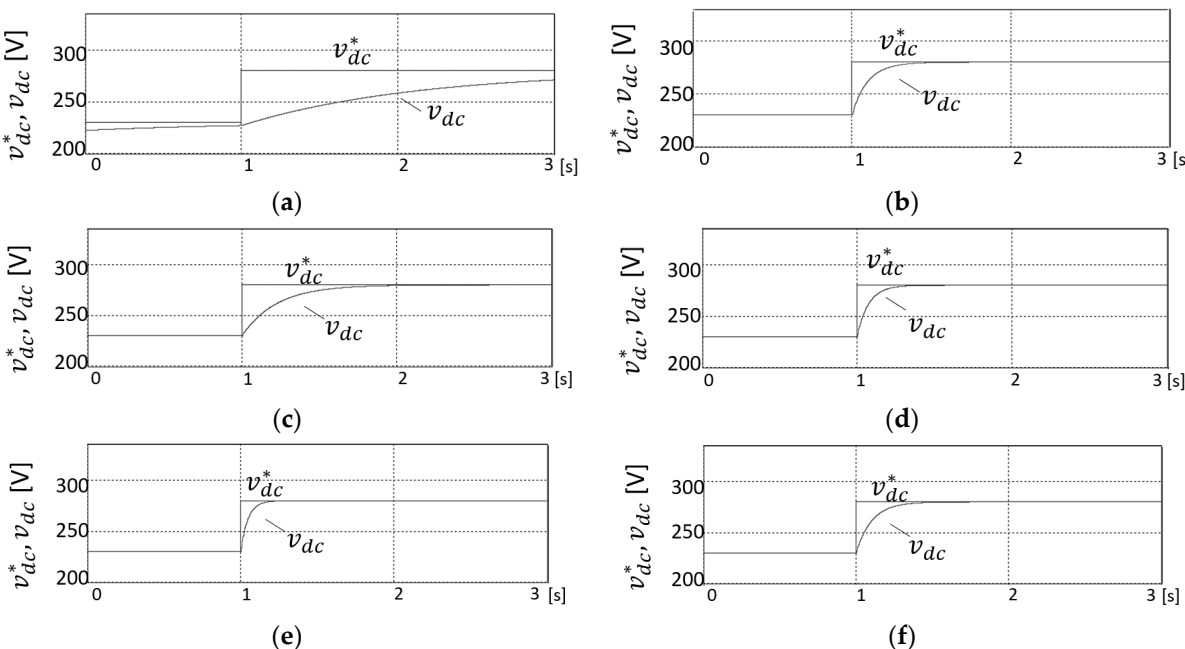

**Figure 10.** Simulated results for weight coefficients design. (**a**) Q = diag([1,1,1,1,1]); (**b**) Q = diag([1,1,1,100,100]); (**c**) Q = diag([5,5,5,100,100]); (**d**) Q = diag([5,5,5,100,1000]); (**e**) Q = diag([5,5,2,100,1000]); (**f**) Q = diag([5,5,10,100,1000]).

**Table 3.** Parameters of LQR for Q = diag [5,5,2,100,1000], R = diag [1,1].

| | Feedback Gain | | |
|---|---|---|---|
| $K_1$ | −0.028 | $K_6$ | 1.310 |
| $K_2$ | −1.060 | $G_1$ | 8.072 |
| $K_3$ | 0.744 | $G_2$ | 0.886 |
| $K_4$ | 0.133 | $G_3$ | −56.322 |
| $K_5$ | 3.302 | $G_4$ | −45.360 |

*5.3. Boost Operation and Neutral Point Potential Control Characteristics*

Figure 11 shows the experimental results when the proposed LQR is used in the circuit shown in Figure 2. The output voltage command value is 280 V and the neutral point potential command value is 0 V. The load resistance is 200 Ω. Figure 11 shows that both the output voltage and neutral point potential follow the command value well.

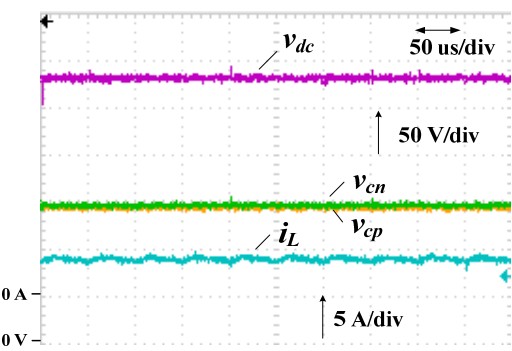

**Figure 11.** Steady state characteristics of series circuit.

*5.4. Input Current Ripple Characteristics*

Figures 12 and 13 show the experimental results of the input current ripple. The load resistance is 100 Ω ($R_p = R_n = 50$ Ω). Figures 12 and 13 show the input current ripple of

the parallel and series circuits, respectively. These are the results obtained by open-loop control for the duty ratios of 0.3, 0.5, and 0.6, respectively. When *D* = 0.3 and 0.6, the input current ripple of the series circuit is approximately one-quarter that of the parallel circuit. Figures 14 and 15 show the simulated results of the input current for the same load condition. It can be seen that the simulation and experimental results are in good agreement for *D* = 0.5 and *D* = 0.6. Comparing Figures 12a and 14a in the case of *D* = 0.3, there is a difference between them. It can be mainly attributed to the effect of the turn on and turn off time in the experiment. Figure 16 shows a comparison of the theoretical, simulated, and experimental results for the input current ripple. In the experimental results, the duty ratio was set to a maximum of 0.65 due to the limitation of the rated voltage of the experiment equipment. As shown in Figure 16, the input current ripple of the series circuit is one-quarter that of the parallel circuit.

*5.5. Output Voltage Response Characteristics*

Figure 17 shows the response of the output voltage and input current when the voltage command is changed in steps from 150 V to 200 V. Both the output voltages in cases Figure 17a,b respond in about 100 ms, which is the desired response. Figure 18 shows the response when the load is changed from 200 W to 500 W. In case Figure 18a, the output voltage drops by a maximum of about 25 V and fluctuates for 140 ms, while in case Figure 18b, the output voltage drops by up to 8 V but recovers in about 10 ms. Figure 19 shows the response when the load is varied from 500 W to 200 W. In case Figure 19a, the output voltage rises up to about 20 V and fluctuates over 140 ms. In case Figure 19b, the output voltage rises by a maximum of about 8 V but recovers in about 10 ms. These results show that the response of the proposed LQR to load fluctuations is superior to that of PI control.

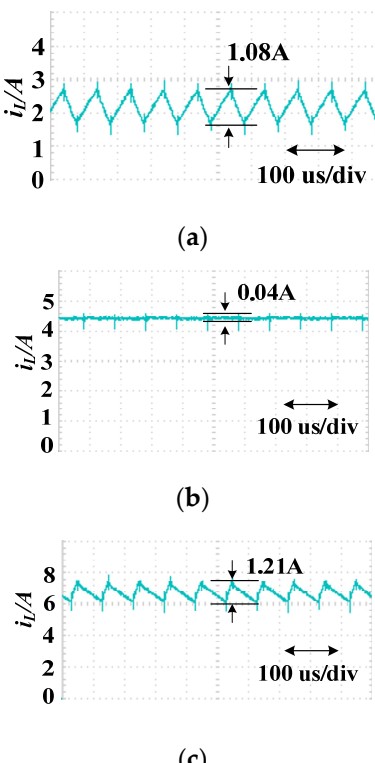

**Figure 12.** Experimental results of input current ripple characteristics of parallel circuit. (**a**) *D* = 0.3; (**b**) *D* = 0.5; (**c**) *D* = 0.6.

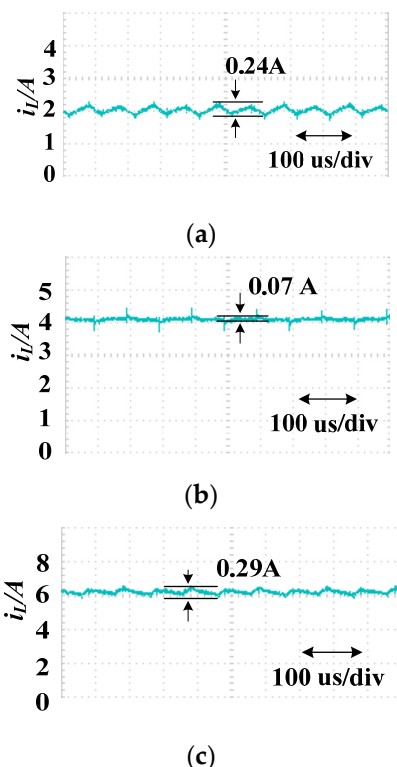

**Figure 13.** Experimental results of input current ripple characteristics of series circuit. (**a**) *D* = 0.3; (**b**) *D* = 0.5; (**c**) *D* = 0.6.

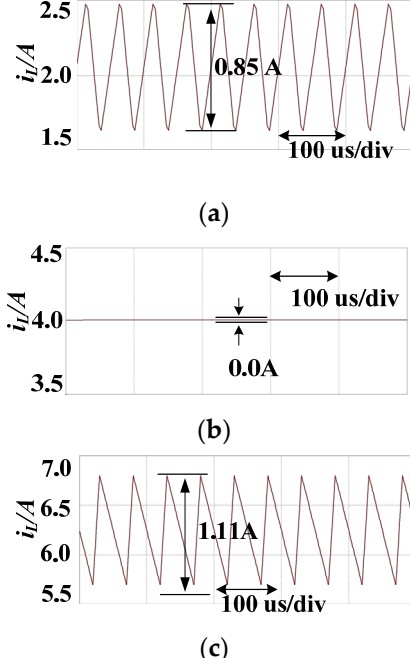

**Figure 14.** Simulated results of input current ripple characteristics of parallel circuit. (**a**) *D*=0.3; (**b**) *D* = 0.5; (**c**) *D* = 0.6.

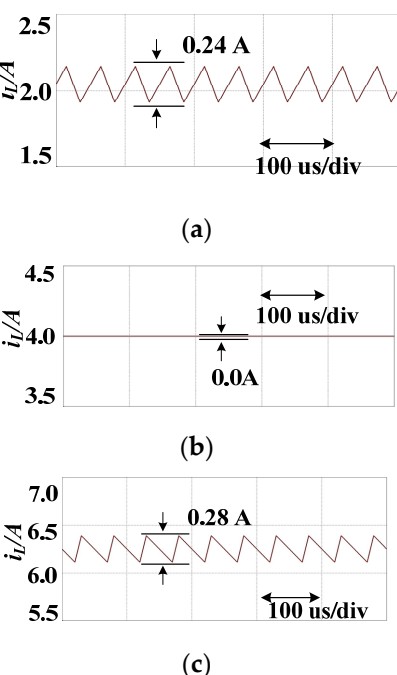

**Figure 15.** Simulated results of input current ripple characteristics of series circuit. (**a**) *D* = 0.3; (**b**) *D* = 0.5; (**c**) *D* = 0.6.

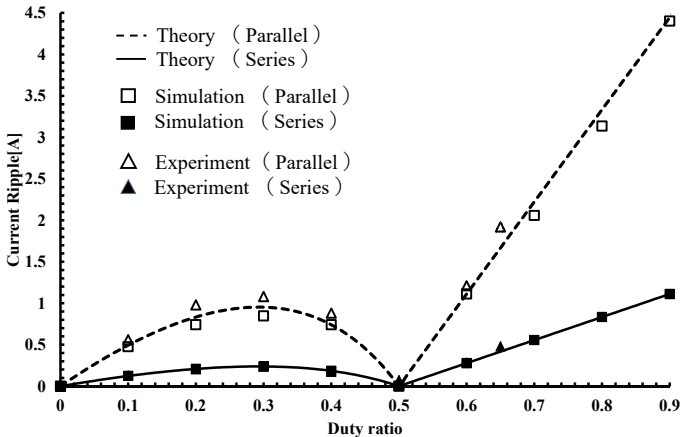

**Figure 16.** Input current ripple characteristics for duty ratio.

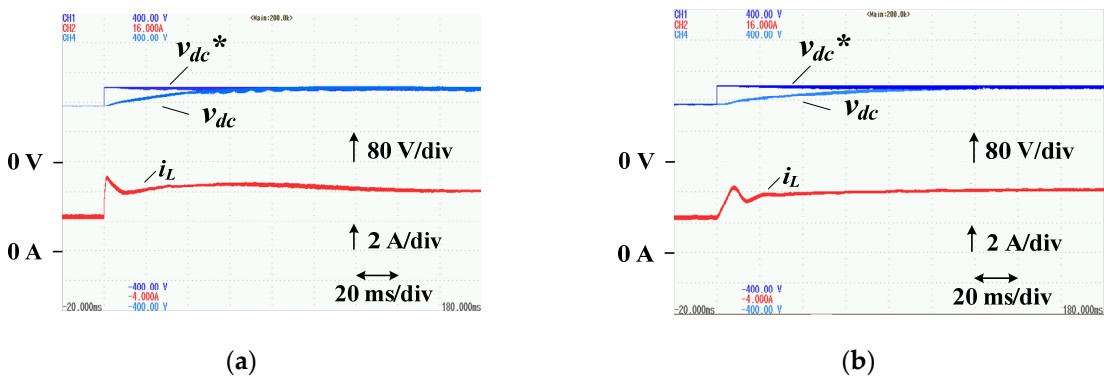

**Figure 17.** Step response (150 V to 200 V). (**a**) PI control; (**b**) LQR.

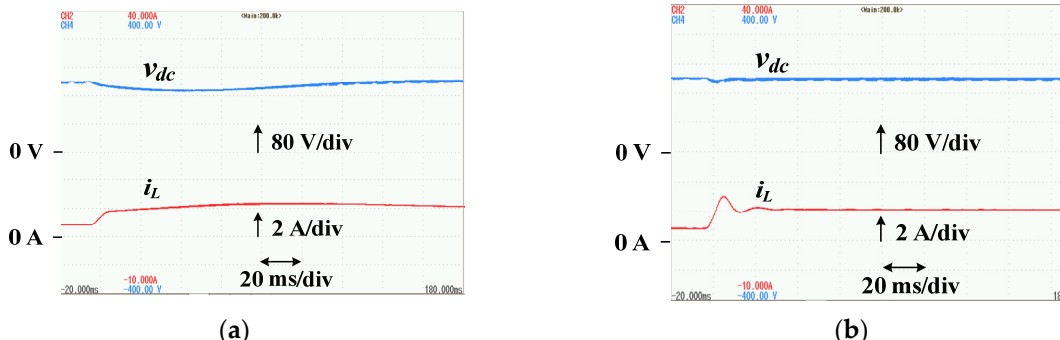

**Figure 18.** Load variation (200 W to 500 W). (**a**) PI control; (**b**) LQR.

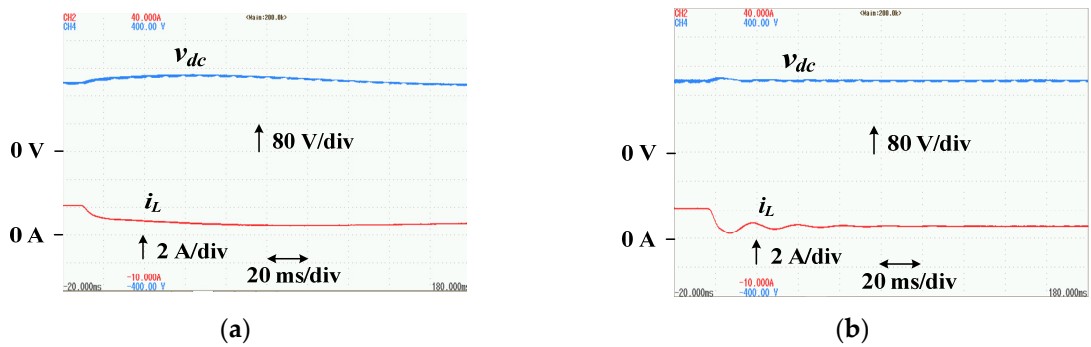

**Figure 19.** Load variation (500 W to 200 W). (**a**) PI control; (**b**) LQR.

*5.6. Efficiency Characteristics*

Figure 20 shows the characteristics of conversion efficiency versus output voltage. When the load power is 500 W, the parallel circuit has high efficiency when the output voltage is 160 V (duty ratio 0.38) or lower. At higher output voltages, the series circuit is highly efficient. When the load power is 800 W, the series circuit is higher in efficiency when the output voltage is 240 V (duty ratio 0.58) or higher. Thus, the conversion efficiency of the series circuit is higher than that of the parallel circuit at light loads and high output voltages.

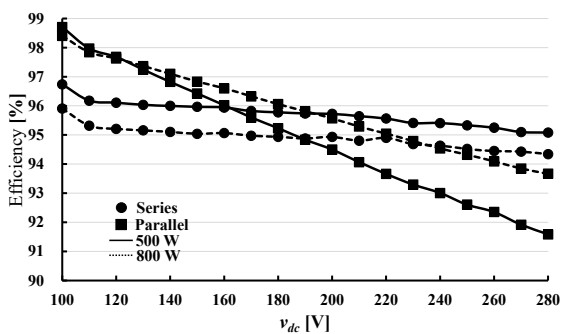

**Figure 20.** Efficiency characteristics for output voltage.

Figure 21 shows a comparison of the efficiency with respect to the load. As for 280 V, the efficiency of the series circuit is higher than that of the parallel circuit over the entire range. As for 200 V, the efficiency of the series circuit is higher below 630 W. When the load power is 300 W, the efficiency of the series circuit is 96%, compared to 92% for the parallel circuit. These results show that the series circuit has an advantage in conversion efficiency in the light load and high voltage range. Figures 22 and 23 show the loss separation results for the parallel and series circuits. When the output power is 500 W, the switching losses are 32.4 W and 12.5 W, and the iron losses are 19.5 W and 5.22 W, respectively. The iron loss

of the series circuit is about one-quarter that of the parallel circuit, and the switching loss is about one-third. The reason of that is related to the input current ripple $\Delta i_L$ described in Section 5.4. The difference between the input current ripple $\Delta i_L$ of the two circuits becomes large when the duty ratio is greater than 0.5. The input current ripple $\Delta i_L$ of the parallel circuit is larger than that of the series circuit. Therefore, as shown in Figure 20, the efficiency of the parallel circuit decreases largely due to increased switching losses.

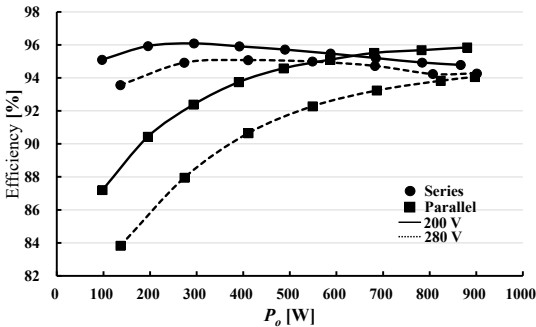

**Figure 21.** Efficiency characteristics for output power.

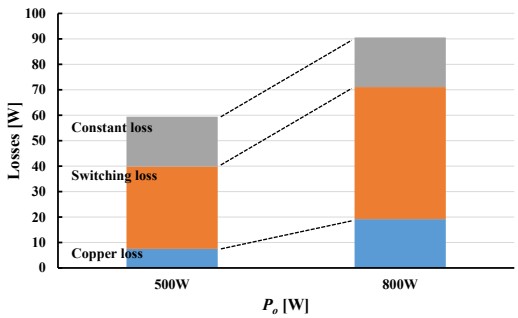

**Figure 22.** Loss comparison of parallel circuit when output voltage is 280 V.

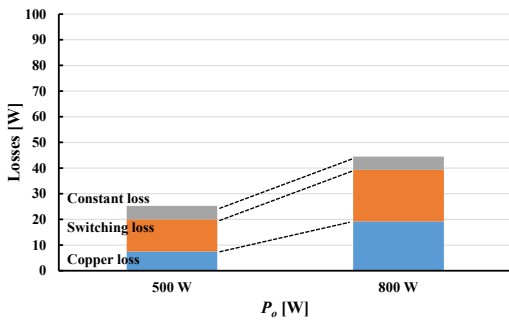

**Figure 23.** Loss comparison of series circuit when output voltage is 280 V.

## 6. Conclusions

In this paper, an interleaving scheme for a series connected boost converter was proposed for application to a three-level inverter. The effectiveness of the proposed interleaving scheme is clarified by comparing the input current ripple characteristics with those of the parallel-connection-type interleaving scheme. The conclusions obtained in this paper are as follows:

i.  The series circuit is capable of both boosting and neutral potential control by means of a boost converter connected in a dependent manner, and is effective for use in a three-level inverter.

ii.  A theoretical analysis of the input current ripple of the series circuit was performed. The validity of the analysis was demonstrated by experiments and simulations using an experimental apparatus with an output voltage of 280 V.

iii.  When the inductances of the series circuit and the parallel circuit are equal, the input current ripple of the series circuit is one-quarter that of the parallel circuit.

iv.  When the inductance of the series circuit is one-quarter of the parallel circuit and the volume and weight of both circuits are equal, the input current ripple of the two circuits are equal.

v.  As an output voltage control method for the series circuit, a control method using an optimal regulator was proposed, and a design method using the state averaging method is presented. The proposed optimal regulator has a better load regulation response than the general PI control method, suggesting that it is effective in downsizing the smoothing capacitor.

vi.  The series circuit has higher conversion efficiency in the light load, high voltage region than the parallel circuit, and is suitable for operation in this region.

**Author Contributions:** Conceptualization, R.C.; methodology, R.C., E.S. and T.S.; software, R.C. and E.S.; validation, E.S. and R.S.; formal analysis, E.S. and R.S.; investigation, E.S., H.H. and R.M.K.; writing—review and editing, E.S., H.H. and R.M.K.; supervision, R.M.K.; project administration, E.S.; funding acquisition, E.S. and H.H. All authors have read and agreed to the published version of the manuscript.

**Funding:** This work was supported by JSPS KAKENHI Grant Number JP21K04038.

**Institutional Review Board Statement:** Not applicable.

**Informed Consent Statement:** Not applicable.

**Data Availability Statement:** Not applicable.

**Conflicts of Interest:** The authors declare no conflict of interest.

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
