# Peer review of "Comparison of Interleaved Boost Converter and Two-Phase Boost Converter Characteristics for Three-Level Inverters"

_wevj, doi:10.3390/wevj14010007_

Round 1

Reviewer 1 Report

1.  The novelty and contribution of the manuscript should be stressed.

2.  The main application of the two-phase boost converter should be introduced in the introduction.

3.  There are some errors in firgures and equations. Please revise them.

4.  Why is the current ripple is so large when the duty ratio is larger than 0.5 in Fig. 16? Please give the reason.

Author Response

Dear Reviewer 1

Thank you so much for your time and for reviewing our manuscript for publication in the Journal of WEVJ. I would like to respond to your comments, point by point.

Sincerely yours,

Eiichi Sakasegawa

Reviewer 2 Report

The article does not explain in detail why a voltage level increasing converter is needed for hybrid transport. Is it a main engine driving converter or a regenerative braking energy converter?

Author Response

Dear Reviewer 2

Thank you so much for your time and for reviewing our manuscript for publication in the Journal of WEVJ. I would like to respond to your comments, point by point.

Sincerely yours,

Eiichi Sakasegawa
